# Mapping enzyme catalysis with metabolic biosensing

Linfeng Xu [1], Kai-Chun Chang[1], Emory M. Payne [2], Cyrus Modavi[1], Leqian Liu[1], Claire M. Palmer [3], Nannan Tao[4], Hal S. Alper [3,5], Robert T. Kennedy [2], Dale S. Cornett[6] & Adam R. Abate [1,7 ✉]

Enzymes are represented across a vast space of protein sequences and structural forms and have activities that far exceed the best chemical catalysts; however, engineering them to have novel or enhanced activity is limited by technologies for sensing product formation. Here, we describe a general and scalable approach for characterizing enzyme activity that uses the metabolism of the host cell as a biosensor by which to infer product formation. Since different products consume different molecules in their synthesis, they perturb host metabolism in unique ways that can be measured by mass spectrometry. This provides a general way by which to sense product formation, to discover unexpected products and map the effects of mutagenesis.

---

[1] Department of Bioengineering and Therapeutic sciences, University of California, San Francisco, San Francisco, CA, USA. [2] Department of Chemistry, University of Michigan, Ann Arbor, MI, USA. [3] Institute for Cellular and Molecular Biology, The University of Texas at Austin, Austin, TX, USA. [4] Bruker Nano Surfaces, San Jose, CA, USA. [5] McKetta Department of Chemical Engineering, The University of Texas at Austin, Austin, TX, USA. [6] Bruker Daltonics, Billerica, MA, USA. [7] Chan Zuckerberg Biohub, San Francisco, CA, USA. ✉email: adam@abatelab.org

Enzyme engineering uses an iterative cycle in which libraries of gene variants are designed, synthesized into proteins, and tested for the activity of interest[1]. The success of these engineering campaigns, however, is dependent on technologies for conducting these steps. Moreover, while there have been significant advances in design and build, the test remains a bottleneck[2]. For example, the dominant strategy is well plate screening, because it is simple and flexible, and allows a variety of measurement techniques to directly quantify the product of the enzyme[2,3]. Well plate screening, however, is severely limited in scalability, testing just hundreds of variants per cycle[3]; this is a major issue because of the likelihood of identifying superior variants scales with the number screened. Thus, alternative approaches based on selections, flow cytometry, and droplet microfluidics are valuable because they afford much higher throughput, screening $>10^7$ variants per cycle[2,4]. However, a major constraint of these approaches is that they do not directly detect product formation, requiring a secondary assay to tether it to a detectable readout[4]. Identifying such assays can be challenging and, often, is not possible for enzymes of interest[2]. To enhance our ability to engineer enzymes through screening, a new method is needed that combines the generality of well plates with the scalability of microfluidics.

In this paper, we describe a screening approach that combines the scalability of microfluidics with the generalizability of mass spectrometry (MS) (Fig. 1). Our approach leverages the background metabolism of the host as a biosensor with which to assess the activity of an enzyme embedded in it, and a µMS technology to characterize metabolic changes. Since the enzyme catalyzes molecules of central metabolism, its activity perturbs the host's metabolite profile[1,5], generating a signature that can be detected even if the enzyme product is not directly observed. Our approach thus provides a general way to map the catalysis of a mutated enzyme, to characterize the range of products it generates, and to recover the sequences of variants with desired activities.

## Results

Matrix-assisted laser desorption ionization (MALDI)-MS has been used for single-cell metabolomics[6] and to identify enzyme products from microbial colonies[7]. Here, we combine this approach with printed droplet microfluidics (PDM) to prepare, print, and screen all mutants from a semi-rationally varied four-position library of the *Gerbera hybrida* G2PS1 type-3 polyketide synthase[8], comprising 1960 codon-shuffled members (Fig. 2a) (see "Methods"). This enzyme is responsible for the biosynthesis of triacetic acid lactone (TAL) through condensation of a starter acetyl-CoA unit with two malonyl-CoA molecules and subsequent cyclization of the triketide chain[8,9]. TAL has been used as a platform precursor for the synthesis of high-value chemicals commonly derived from fossil fuels[9]. Mutations in the active site of these enzymes can alter the kinetics and spectrum of polyketide products formed[8,10], potentially accessing novel products (Fig. 2b). We synthesize our library into a plasmid backbone (Supplementary Fig. 1) and transform the library into *Yarrowia lipolytica*, encapsulating and culturing single cells in 300 pL droplets to generate isogenic colonies (Supplementary Fig. 2). Culture expansion produces additional material compared to a single cell, boosting MS signal and providing accurate metabolomic data.

**Mapping the catalytic activity of an enzyme variant library with metabolic biosensing.** To perform microscale MS (µMS), we use a high-density plate comprising a glass slide etched with 10,000 wells, having 80 µm diameters and rounded bottoms (Fig. 3a); this shape concentrates the material to the center (Supplementary Fig. 3), enabling accurate µMS quantitation. Higher capacity slides with the dimensions of a MALDI plate can accommodate 100,000 wells (Supplementary Fig. 4). To maximize throughput, all wells must be loaded with one colony, which we accomplish with PDM[11] in ~30 min (Supplementary Fig. 5). During printing, we scan the colonies, dispensing only ones falling within a narrow cell density range (see "Methods")[11].

Using this system, we print 9000 library members and 1000 reference strains to designated positions. Once loaded, the plate is dried, spray-coated with matrix, and subjected to MALDI-MS imaging (see "Methods"), providing data on all detectable metabolites from mass-to-charge ratio ($m/z$) 30 to 630. The results are reported as an "image" in which each pixel comprises a 600-dimensional vector of signal amplitude for each $m/z$ (Supplementary Figs. 6–8 and Supplementary Data 1). Thus, the data can be thought of as representing a 600 "color" image, in which each slice reports the amplitude of the metabolite corresponding to the respective $m/z$. Importantly, MALDI's soft ionization preserves DNA[12], allowing PCR recovery of enzyme genes after MS analysis (Supplementary Fig. 9).

**Metabolic biosensing detects variants with different function.** A valuable property of µMS is that it provides information on many molecules in the host cells, enabling the discovery of unexpected enzyme activities. A general way to identify activities is to plot alterations in the cell metabolome as a Uniform Manifold Approximation and Projection (UMAP), a dimensionality reduction technique used in single-cell sequencing[13,14] and MS imaging[15] that projects high-dimensional data onto a plane while preserving cluster information. To obtain the clearest UMAP clustering, we apply an algorithm to select the best $m/z$

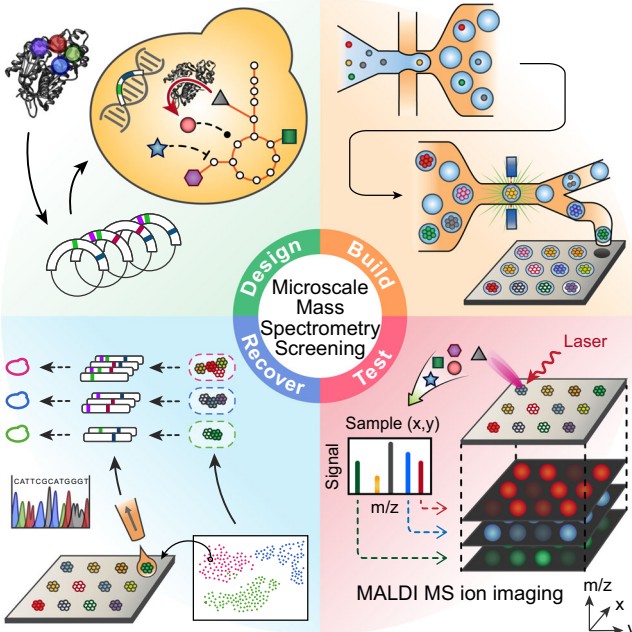

**Fig. 1 Metabolic biosensing with microscale mass spectrometry provides a general strategy for screening enzymes.** Enzyme variants are designed and transformed into yeast (design) and then synthesized in the yeast where they consume molecules of central metabolism to generate product (build). Using printed droplet microfluidics, they are dispensed to a picoliter well array and subjected to MALDI-MS imaging to quantify cell metabolites (test). UMAP clusters cells according to metabolomic profile, where each cluster indicates a different enzyme phenotype. Desired mutants are extracted from the plate, sequenced, and confirmed in bulk cultures.

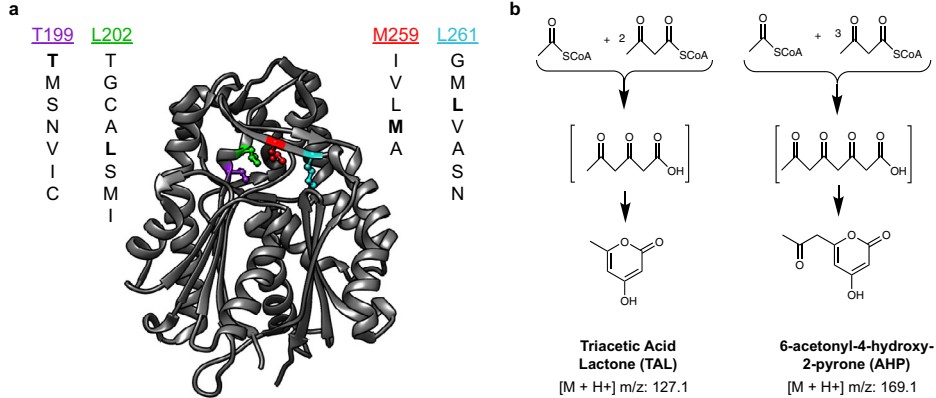

**Fig. 2 Overview of the type-3 PKS variant library. a** Crystal structure of *Gerbera hybrida* G2PS1 (PDB ID: 1EE0) showing the location of the four residues selected for mutagenesis and identity of the residue mutants. All residues except T199 directly form the active site cavity. **b** The smallest condensation/cyclization products expected from type III polyketide synthase activity: triacetic acid lactone (TAL, the native product of G2PS1) from one acetyl-CoA and two malonyl-CoA and 6-acetyl-4-hydroxy-2-pyrone (AHP) from one acetyl-CoA and three malonyl-CoA. Higher-order polyketides, not shown, are possible from additional condensations of malonyl-CoA.

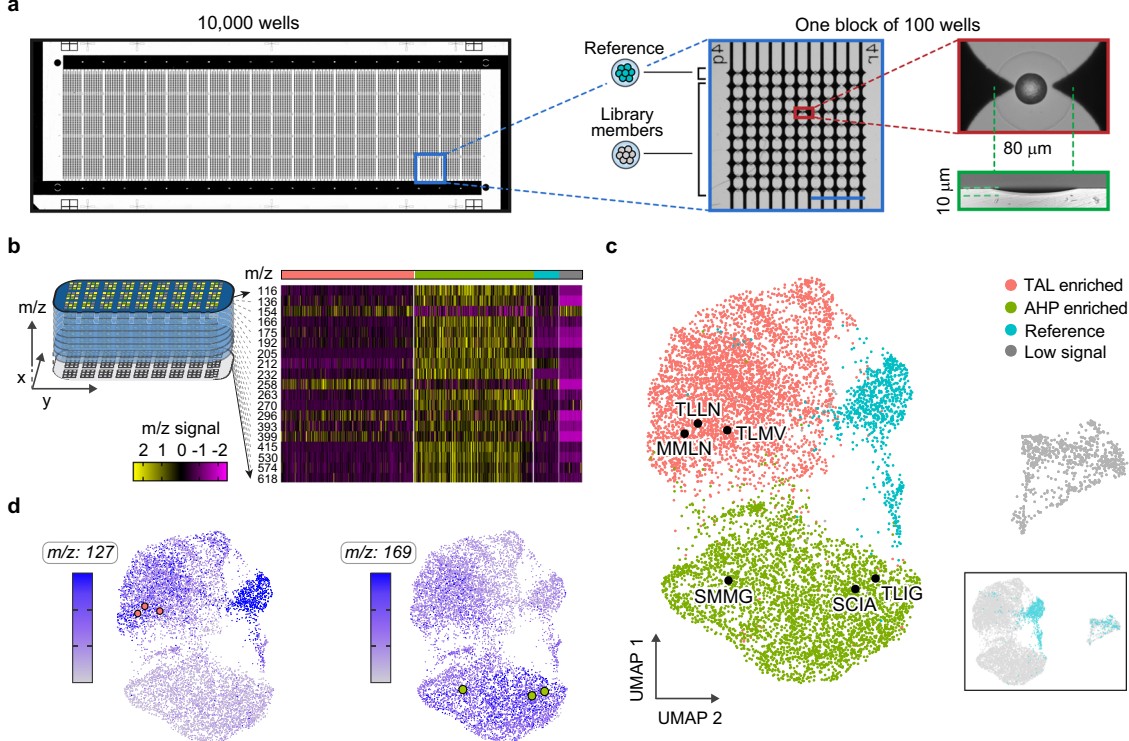

**Fig. 3 Mapping g2ps1 mutant catalysis through metabolic biosensing. a** Overview of the slide (75 mm × 25 mm) for single variant μMS comprising 10,000 wells grouped in 100 blocks of 100 wells (magnified blue box), scale bar 1 mm. Each circular well is paired with electrodes for printed droplet trapping (red box) and has a rounded profile that concentrates desiccated metabolites to the center (green). **b** μMS generates a spatial image of the substrate in which each pixel contains a full *m/z* spectrum. **c** Well spectra are extracted and clustered into four groups using UMAP. Mutants of interest are recovered, sequenced, and overlaid on the clusters as black dots. Inset shows the reference strain based on known print locations. **d** Heat maps for *m/z* 127 and 189 indicate high production of TAL in the upper and reference clusters, and high production of AHP in the lower cluster. Source data are provided in Supplementary Data 1.

peaks for inclusion, comprising ~60% of the total *m/z* intensity (Fig. 3b, Supplementary Figs. 9 and 10, and Supplementary Movies 1 and 2) (see "Methods"). From this, we observe four clusters (Fig. 3c) that, visually, resemble UMAP clusters of single-cell RNA-sequencing profiles, except that they correspond to metabolite profiles[7,16]. The compact blue cluster is the 1000 control reference strains, which we confirm by substrate location (Fig. 3c, inset). The gray island corresponds to wells with few cells

that were misprinted (Supplementary Fig. 11). This leaves two large clusters (red and green) that, presumably, correspond to cells with distinct metabolomic profiles. Since only the embedded enzyme varies between these cells, this implies the codon-shuffled library exhibits two major activities that perturb cell metabolism in distinct ways. Mapping the *m/z* 127 data onto the clusters shows that the upper red island represents productive TAL mutants (Fig. 3d, left UMAP). In addition to the desired product,

**a**

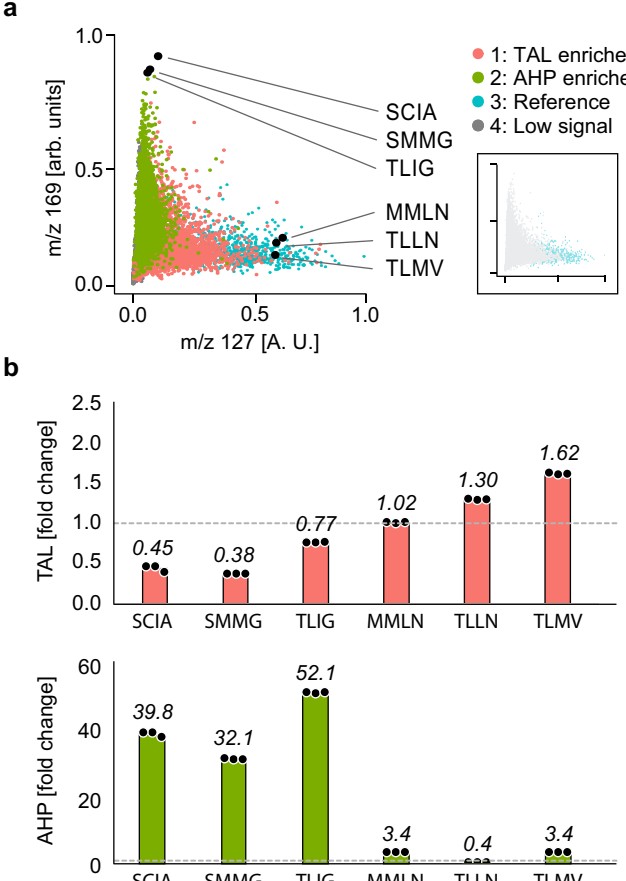

**b**

**Fig. 4 µMS screen of g2ps1 mutant library for increased production of TAL or AHP. a** Normalized *m/z* intensity for *m/z* 127 (TAL) versus 169 (AHP). Data points are colored based on clusters identified from UMAP analysis in Fig. 2. Inset shows reference strain samples. The highest producers for TAL and AHP were isolated, sequenced, and confirmed in bulk analysis (black circles). **b** TAL and AHP for selected bulk cultures as fold change over wild type (TLML). $n = 3$ independent experiments. Bar plots represent mean values, with black dots representing each measurement of the samples. Source data are provided as Source Data and raw data in Supplementary Data 5 and 6.

some mutants generate another product dominantly produced in the lower green cluster. Upon careful inspection of the MS data, we determine that a molecule with the *m/z* value of 169 has differential abundance between these clusters (Supplementary Fig. 12 and Supplementary Data 1 and 2). Using high-performance liquid chromatography (HPLC) with tandem MS (HPLC-MS/MS), we confirm that the mass corresponds to the reported alternative product of this enzyme, tetraketide 6-acet-onyl-4-hydroxy-2-pyrone (AHP)[10,17]. When we overlay the AHP amplitude (*m/z* 169) on the UMAP, it is predominantly found in the green cluster (Fig. 3d, right UMAP).

To characterize how these disparate activities relate to enzyme sequence, we select and sequence ~50 mutants from each cluster (Supplementary Data 3 and 4). Plotting the mutants versus normalized TAL and AHP shows colonies with extreme activity and, generally, that mutants efficient at producing one product are inefficient at producing the other (Fig. 4a). To validate these results, we re-transform and test several mutants in bulk, finding good agreement (Fig. 4b, Supplementary Fig. 13, and Supplementary Data 3). Interestingly, the TLLL mutant based on consensus design[18] of TAL-enriched members from the UMAP

also shows high TAL production, although it retains higher AHP side activity than the TLLN mutant (Supplementary Fig. 14). These results demonstrate that the generation of these products can be inferred from clustering the host cells' metabolomic profiles, even though the peaks associated with these molecules (*m/z* 127 and 169) are not included in the clustering. In this way, the host cell's metabolite profile affords a biosensor by which to infer changes in the embedded enzyme's activity.

## Discussion

Metabolic biosensing allows inference of product formation even without direct product detection because the enzyme's activity perturbs the host cell's metabolite profile. Provided such perturbations are distinct for each activity, all activities generated by a mutant library can be detected, similar to the "compressed sensing" of smell sensation[19]. Such indirect sensing has beneficial features for enzyme engineering. In addition to providing a universal readout of product formation, the metabolites used to generate the UMAP can be selected to achieve the best cluster differentiation, even if the direct enzyme products are not included because they are difficult to detect due to their chemical or ionization properties. Furthermore, it should allow detection of all possible products, since each activity should perturb the host metabolite profile in unique ways and, thus, generate distinct clusters in the UMAP. By subsampling the clusters, unexpected products can be discovered. While we have utilized this analysis approach with yeast cells and MALDI-MS, it should apply to other bioproduction systems, including cell-free extracts, bacteria, and mammalian cells. Other readout modalities that provide sensitive, multiplexed detection of metabolites should be applicable, including other forms of mass spectrometry[20] and multi-parametric spectroscopy[21]. Lastly, while we have focused on an enzyme library for its simplicity, the approach should apply to any engineering that perturbs host cell metabolism, including genetic circuits and biosynthetic pathways.

## Methods

**Yeast library design and construction.** Residues in the binding pocket, T199, L202, M259, and L261, of 2-pyrone synthase (2PS) that determine substrate specificity and/or control product size are selected based on sequence homology as described[8]. To recombine homologous mutations, these residues are combinatorically mutated to either the ones described previously[8,9,22] or the naturally and frequently occurring ones calculated from the position-specific scoring matrix using the ConSurf server (https://consurf.tau.ac.il/) and PSI-BLAST. We included the mutations of T, M, S, N, V, I, and C for T199; T, G, C, A, L, S, M, and I for L202; I, V, L, M, and A for M259; and G, M, L, V, A, S, and N for L261. The combinations of these variants sum to a library size of 1960.

*Yarrowia lipolytica* is chosen as the expression organism due to its high tolerance to salts, TAL[9], and many other chemicals[23], and its robustness in droplet culture. The high lipid flux in this oleaginous organism also suggests a high flux of acetyl-CoA[9], from which TAL is derived.

Codon-optimized wild-type *g2ps1* gene is purchased from Addgene (pJY3872, Addgene plasmid # 113338, Supplementary Data 7)[24]. The mutational library is synthesized based on the pJY3872 plasmid by GenScript combinatorial mutagenesis service with equal distribution of each amino acid mutation. Upon receipt, the mutated region of the library is amplified by PCR for subsequent cloning (lib-for primer: GAG CAC ATG GTG GAC CTG and lib-rev primer: CTC CTT AAG GTT CAG CTT ACG C). To integrate the mutant genes into *Y. lipolytica*, both the *Y. lipolytica* Po1g-competent cells and the expression vector pYLEX1 are purchased from Yeastern Biotech. The pYLEX1 vector is linearized by *Kpn*I and *Pml*I (R0142S and R0532S, New England Biolabs) digestion. The wild-type *g2ps1* gene is attached with cloning adaptors by PCR (2ps-for primer: AAC CAC ACA CAT CCA CAA TGG GCT CTT ACT CCT CTG AC and 2ps-rev primer: CAG GCC ATG GAG GTA CTC AAT TAC CGT TGG CCA C), and then cloned into linearized pYLEX1 using the Seamless Cloning Kit (Yeastern Biotech) to create the wild-type expression plasmid pYLEX-2PS. To increase 2PS expression yield and improve detection for potential products from 2PS, the synthetic promoter UAS1B16-TEF(504)[25] is purchased from Addgene (pUC-UAS1B16-TEF(504), Addgene plasmid # 44371). The 16 tandem copies of upstream activation sequences (UASs) are expected to increase messenger RNA expression by ~350-folds relative to the basal promoter[25]. The restriction sites of *Bst*B1 and *Asc*I are inserted upstream and downstream of the hp4d synthetic promoter of

pYLEX-2PS by PCR (vecAscI-for primer: TTG GCG CGC CAT GGG CTC TTA CTC CTC TG and vecBstB1-rev primer: CTG CAG AAT TCG AAC CTC AGC ATG CAC CAT TC), respectively. The UAS1B16-TEF(504) promoter is subsequently cloned into pYLEX-2PS to replace the hp4d promoter and create the high-expression plasmid pTEF-2PS by restriction enzyme digestion and ligation (Supplementary Fig. 1). The pTEF-2PS plasmid is linearized by *Dra*III and *Afl*II (R3510S and R0520S, New England Biolabs) digestion. The library amplicons are cloned into linearized pTEF-2PS by NEBuilder HiFi DNA Assembly (E2621X, New England Biolabs). All cloning plasmids are propagated in NEB 5-alpha-competent *Escherichia coli* cells (C2987H, New England Biolabs) as per the manufacturer's instructions.

After transformation of the library into *E. coli*, ~14,000 colonies are counted and pooled from LB-ampicillin selection plates (L5667, Sigma). Transformants resulting from incomplete pTEF-2PS digestion are estimated to account for 6% of the population, as calculated from negative controls without the mutant inserts. Pooled *E. coli* colonies are cultured in LB-ampicillin broth at 37 °C with shaking overnight, followed by plasmid MiniPrep. The plasmid library is linearized by *Not*I (R0189S, New England Biolabs) digestion to allow genomic integration into the pBR322 docking platform of *Y. lipolytica*. Transformation and leucine selection are carried out as per the manufacturer's instructions. The yeast colonies are randomly picked for PCR and Sanger sequencing to ensure successful genomic integration. A total of ~2400 colonies are recovered and pooled from SD-Leu selection plates (630311, Takara) for subsequent liquid culture in SD-Leu broth at 30 °C with shaking. The library complexity is validated by amplifying the mutated region with PCR (Read2-2PS-for primer: GTC TCG TGG GCT CGG AGA TGT GTA TAA GAG ACA GTG ATC GTG TGC TCG GAG ATC and Read1-2PS-rev primer: TCG TCG GCA GCG TCA GAT GTG TAT AAG AGA CAG GGT TAG ACC TCC TTC TCT CAG GTG), followed by amplicon NGS with 500-cycle MiSeq Reagent Nano Kit v2 (MS-102-2003, Illumina).

**Optimized codon sequence for the *g2ps1* gene.** From Addgene: pJY3948[24]: ATGGGCTCTTACTCCTCTGACGATGTGGAGGTGATCCGAGAGGCTGGAC GAGCCCAAGGCCTAGCTACCATCCTCGCCATTGGCACTGCAACCCCTC CCAACTGTGTCGCTCAGGCCGATTACGCCGACTACTACTTTCGGGTCA CGAAGTCTGAGCACATGGTGGACCTGAAGGAGAAGTTCAAGCGGATTT GTGAGAAAACCGCCATCAAAAAGCGATACCTGGCCCTGACCGAGGAC TACCTGCAAGAGAACCCTACCATGTGTGAGTTCATGGCACCTTCCCTG AACGCCCGACAGGACCTGGTGGTGACAGGAGTGCCTATGCTGGGAAA GGAGGCTGCCGTGAAGGCCATTGATGAGTGGGGCTTGCCGAAGTCTAAG ATCACCCATCTGATCTTCTGCACCACTGCCGGAGTTGACATGCCTGGAG CCGATTATCAGCTGGTGAAGCTACTGGGACTGTCTCCCTCTGTGAAGC GGTACATGCTGTACCAACAGGGTTGTGCCGCTGGAGGTACAGTCCTGCG ACTGGCTAAGGACCTCGCTGAGAACAACAAAGGATCCAGAGTGCTGAT CGTGTGTGCTCGGAGATCACTGCTATCCTGTTTCACGGACCTAACGAGAAC CACCTGCACTCCCTCGTTGCTCAAGCTCTCTTCGGGAGACGGAGCAGCA GCATTGATCGTGGGATCTGGTCCTCACTTGGCAGTTGAACGGCCCATC TTCGAGATCGTGTCTACCGACCAAACCATCCTGCCTGACACCGAGAAAG CCATGAAGCTGCACCTGAGAGAAGGAGGTCTAACCTTCCAACTCCATC GAGATGTGCCCCTTATGGTCGCTAAGAACATCGAGAACGCTGCAGAGA AGGCTTTGAGCCCGCTTTGCATCACCGATTGGAACTCCGTGTTCTGGAT GGTGCATCCCGGAGGTCGAGCCATCCTGGACCAAGTGGAGCGTAAGCTG AACCTTAAGGAGGACAAGCTGAGAGCCTCCAGACACGTTTTGTCTGAG TATGGCAACCTGATCTCTGCCTGTGTGCTGTTCATCATCGACGAAGTGC GAAAGCGTAGCATGGCTGAGGGAAAGTCCACTACGGGAGAGGGACTGG ATTGTGGTGTCCTGTTCGGATTTGGCCCCGGAATGACTGTTGAGACTGTC GTGCTGAGATCCGTGCGAGTCACTGCAGCTGTGGCCAACGGTAATTGA.

**Microfluidic device fabrication.** All the microfluidic devices including the printer head, droplet maker, and droplet merger are made from poly(dimethylsiloxane) (PDMS) based on the protocols of standard soft lithography[26]. For the print head, a two-layer SU 8 master mold is made. The first SU 8 layer, the flow channel layer for droplet reinjection and sorting, is made to 80-μm-thick and a second layer of 20-μm-thick SU 8 is added on top of the first layer only at regions that comprise the channels used to guide insertion of the nozzle and optical fibers. After the master mold is made, uncured PDMS (10:1 polymer to cross-linker ratio) is poured onto the master and cured in an oven at 65 °C for 1 h. The cured PDMS slab is peeled off and ports are punched using a 0.75-mm biopsy core. The PDMS casting is plasma bonded to a 25 mm × 75 mm glass slide and baked at 65 °C overnight. One centimeter of PE/5 tubing (BB31695-PE/5, Scientific Commodities) is inserted into the output channel of the print head and serves as the printing nozzle. The assembled printing device is then treated with AquaPel (Item: 47100, AquaPel) and air-dried. Similarly, the droplet encapsulation device and droplet merger are made from PDMS with a channel height of 70 μm.

**Well substrate fabrication.** The well substrate is fabricated on a standard 25 mm × 75 mm glass slide. The array consists of 10,000 wells arranged in a 200 by 50 format. The diameter and the pitch of the wells are 100 and 250 μm, respectively. Below each well are two sawtooth electrodes pointing to each other with a 30 μm gap in between. To make such a well substrate, we deposit a 200-angstrom-

thick chromium layer on one side of the glass slide by electron sputtering (Huifong Company). Then by photolithography, sawtooth-shaped electrode patterns are transferred from the photomask to a layer of 2-μm-thick MA-P 1215 positive photoresist (Micro Resist Technology). The glass slide is immersed in Chromium Etchant (651826-500ML, Sigma) and cleaned thoroughly by acetone, isopropanol, and deionized (DI) water. After the electrode is made, a 5-μm-thick layer of SU-8 3005 is applied on top of the glass slide, baked, ultraviolet (UV) exposed, and developed. A second layer of 15-μm-thick SU 8 3025 is spun on top and baked pre-exposure for 15 min at 95 °C. Then, by photolithography (UV power at 200 mJ/cm²) and aligning the photomask with fabricated electrodes, patterns of wells are transferred from the photomask to the second layer of SU 8. The coated slide is baked postexposure at 150 °C for 20 min and developed in propylene glycol methyl ether acetate developer (484431-1L, Sigma) for 5 min. The timing and power of UV exposure and postexposure bake are critical to form the bowl-shape profile of the wells, as the shape of the wells is formed by diffusion of moieties from the UV-exposed region of SU 8 to the unexposed parts. The final step is to bake the developed slide again at 95 °C overnight to ensure SU 8 crosslinking. All baking is done on a hot plate.

**Yeast preparation.** Engineered yeast strains (*Y. lipolytica*) with varied copy numbers of *g2ps1* as the reference strain used in this study have been reported[9]. Both yeasts of the reference and library strains are from a frozen glycerol stock. They are first inoculated in a culture medium with 2 mL culture medium (20 g/L glucose, 6.7 g/L YNB with ammonium sulfate, and 0.79 g/L CSM) for one day at 30 °C. The yeasts are then pelleted and resuspended with a fresh culture medium. The yeasts are counted and diluted to $3 \times 10^5$ cells/mL with a fresh culture medium. This concentration is chosen to ensure roughly one in ten droplets contain a single yeast cell when encapsulated.

**Yeast encapsulation and droplet culturing.** Yeast cells in media are diluted to a limiting concentration that yields one encapsulated cell per ten microfluidic droplets. The diluted cell solution and Novec HFE-7500 oil (3M) with 2% PEG–PFPE amphiphilic block copolymer surfactant (008-Fluoro-surfactant, RAN Bio-technologies) are co-flowed in a cross-junction drop maker to generate 80 μm water-in-oil droplets. All droplets are collected into a 5 mL syringe (BD) positioned vertically in a shaking 30 °C incubator for 5 days to form isogenic microcolonies within droplets. The droplets containing the isogenic microcolonies are then ready for printing.

**Print head setup and droplet sorting.** A similar optical setup of the print head from our previous study[11] is used here: a multimode excitation fiber with a core diameter of 105 μm and a numerical aperture of 0.10 (Thorlabs) is inserted into a guide channel in the print head. Similarly, an emission detection fiber with a core diameter of 105 μm and a numerical aperture of 0.22 (Thorlabs) is inserted into a second guide channel in the print head. Four 50 mW continuous-wave lasers with wavelengths of 405, 473, 532, and 640 nm are combined and coupled to the excitation fiber. The emitted light is collimated and ported into a quad-bandpass filter, and then passed through dichroic mirrors. Light passes through bandpass filters of 448, 510, 571, and 697 nm and associated dichroic mirrors to obtain wavelength-specific detection of emitted light with four PMTs. Electrode channels and a "Faraday moat" are filled with a 5 M NaCl solution. A positive electrode is connected to a function generator and a high voltage amplifier while a second electrode is grounded. Fluidic inputs into the PDM device are driven by syringe pumps (New Era). Bias and spacer oil containing Novec HFE-7500 oil are flowed through the device at a flow rate of 2000 μL/h. A waste channel is driven with a negative flow rate of −3000 μL/h. Droplets with yeast cells prepared previously are reinjected into the device at a flow rate of 100 ± 50 μL/h. Real-time optical signal acquisition through a field-programmable gate array (National Instruments) is displayed on custom LabView software. The optical signal is processed in real time and displayed, so droplets of interest (in this case, with similar numbers of yeast cells inside) can be identified by specifying gates. Controlled by our LabView software, droplets are subsequently sorted by passing a high-frequency pulse through a high voltage amplifier (Trek 690E-6). Typical droplet sorting parameters range from 10 to 20 kHz, 50 to 100 cycles, and 0.5 to 1.0 kV.

**Well substrate setup.** The well substrate is immersed in a bath of Novec HFE-7500 oil during the printing operation. Copper tape with a conductive adhesive (16067-1, Ted Pella) is affixed to two electrode contact pads on the well slide. One pad is connected to the ground, while the other is connected to a function generator and a high voltage amplifier, providing an electric field at 200–600 V at 20–30 kHz to trap the sorted droplets in the wells by a dielectrophoresis force.

**Printing procedure.** During the printing process, the print head is fixed to an XYZ micromanipulator and the well substrate is held on a motorized XY mechanical stage (MA-2000, ASI). The printing process is automated by custom LabView software, which coordinates the droplet sorting of the print head and movement of the mechanical stage where the well substrate is held. When printing, the nozzle of the print head is positioned close to the well substrate by the XYZ micromanipulator and the substrate is moved from one well to the next after the desired

droplets are printed. Once printing is complete, the bath oil is removed and the substrate air-dried. The substrate is stored in a petri dish sealed with parafilm at −20 °C until needed.

**Matrix deposition and MALDI-MS imaging**. A matrix solution of 2,5-dihydroxybenzoic acid (DHB) is prepared by adding 15 mg/mL DHB into a solution of 90% acetonitrile and 0.1% trifluoroacetic acid in DI water. It is loaded into an automated matrix sprayer (TM Sprayer, HTX Imaging) that coats the printed well substrate with a layer of DHB matrix at 60 °C using a flow rate of 0.125 mL/min and velocity of 1200 mm/min. Approximately 2 mL of matrix solution is used for each substrate. For MALDI-MS imaging, the rapifleX TOF/TOF (Bruker, Germany) is used in positive ion mode at 50% laser power (4 μJ) and 50 μm pitch.

**MS ion imaging analysis**. A Bruker flexImaging, SciLs Lab software (Bruker, Germany), and Cardinal 2 package in R[27] are used to analyze the MS imaging of the well substrates. The raw MALDI-MS data are processed by Bruker flexImaging to remove background noise, where the signal-to-noise ratio threshold is defined as the height of the mass peak above its baseline relative to three times the standard deviation of the noise[28]. The locations of target wells are identified automatically by a customized MATLAB program. The UMAP analysis is performed by R using the Seurat package[29]. Specifically, the imzML MS imaging file is parsed and visualized with Datacube Explorer, and the mass slice image of the TAL peak (summed over $m/z$ 127.0–127.6) is exported. To locate the printed yeast colonies in the slice image, local TAL intensity maxima are identified by custom Matlab scripts. Next, we run the Cardinal 2 package in R to extract the spectrum of each yeast colony from the imzML file based on the colony coordinates. Spectra from all colonies are normalized by total ion count[30] and aligned using the normalize and mzAlign functions in Cardinal, respectively. The processed spectra are exported to Matlab and binned with a bin size of 1 $m/z$.

To account for instrument drift, the reference strain is printed in every tenth row on the 50 × 200 well array, forming five row "blocks," with the first row of each block being the reference strain. For each row of the reference strain, every $m/z$ peak is averaged over the reference colonies with the top 20% intensities (40 colonies for each row). For the library strains in each row block, every $m/z$ peak is normalized to the corresponding reference intensity from the first row. Top TAL and AHP production colonies are ranked and picked for PCR accordingly.

To identify differential expression of metabolites resulting from 2PS mutations, each $m/z$ peak of the library strains and the reference strain are averaged over the colonies with the top 20% intensities. Next, the difference spectrum is calculated by subtracting the averaged $m/z$ intensities of the reference strain from those of the library strains. Positive peaks in the difference spectrum are thresholded and extracted with the minimum peak distance set to 2 $m/z$ (to include isotopes). The filter threshold is iteratively relaxed (to 0.8 a.u.) to incrementally include more $m/z$ peaks until the library strain can be separated into distinct clusters, yielding 19 $m/z$ peaks with the largest library-reference differences. UMAP clustering of the 19 peaks is conducted with Seurat v4.0 in R without additional dimension reduction. We also tested UMAP clustering using all the MS peaks from the complete dataset, plotted as a three-dimensional (3D) UMAP (Supplementary Fig. 10). We obtain similar clusters to the two-dimensional plot (Supplementary Fig. 10 and Supplementary Movies 1 and 2), although with poorer separation (Fig. 3c). Interestingly, in the 3D plot, we observe a small group emerging from the TAL cluster (Supplementary Fig. 10c, d). Using a volcano plot, we determine that the new cluster expresses a metabolite at $m/z$ 103 of unknown identity. Using the Yeast Metabolome Database[31], we find 19 molecules consistent with $m/z$ 103, although none are known products of the 2PS enzyme. Moreover, this metabolite may not be a direct product of the enzyme, but rather an endogenous metabolite upregulated as a result of the enzyme activity. Metabolite 103 also appears to be highly expressed in the AHP annotated strains at a similar portion in the cluster (Supplementary Fig. 10g).

**Gene recovery from target wells**. A microliquid handler with a 100-μm ID glass micropipette (MPP200B, Bulldog Bio) is used to recover the yeast cells from each of the target wells and transfer the recovered cells to individual PCR tubes (Fisher Scientific). Primers targeting the g2ps1 gene (5′-ATG GGT TCG TAT TCG TCT GA and 3′-TGA CTC GAA CGG ATC GC from IDT) and PCR reagents (Phire Plant Direct PCR Kit, Thermo Fisher) are added into each of the PCR tubes containing the recovered yeast following the protocols of the kit manufacturer and PCR is performed. The amplicon sizes are verified by gel electrophoresis and the sequences are determined by Sanger sequencing (Quintarabio and Genewiz).

**HPLC-MS/MS verification of TAL and AHP production**. For the verification of TAL and AHP production from bulk yeast strain cultures of interest, we generate yeast strains carrying the enzyme variants of interest, culture them in bulk, and perform HPLC-MS/MS analysis. We synthesize the recovered sequences of interest using IDT gBlocks (Supplementary Data 2 and Supplementary Data 5 and 6) and generate the yeast strains in the same method as described in the yeast library construction section. Each yeast strain is cultured in 2 mL of media (20 g/L glucose, 6.7 g/L YNB with ammonium sulfate, and 0.79 g/L CSM-LEU) for 3 days at 30 °C.

Saturated cultures are diluted by 200× into a fresh volume of 2 mL and cultured for another 3 days. Samples of media from each strain are prepared by centrifugation and filtration of cell culture supernatant. For TAL quantification, the supernatant is diluted 1000× in water. For AHP analysis, the supernatant is diluted 100× in water. Samples are quantified using an Agilent 1290 Infinity II outfitted with a Phenomenex Kinetex C18 column (2.1 mm × 100 mm, 1.7 μm) interfaced to an Agilent 6410 triple quadrupole mass spectrometer. The mobile phase is ramped from 5 to 95% acetonitrile in water with 0.1% formic acid over 4 min. The column is re-equilibrated at 5% acetonitrile for 1.5 min. The mass spectrometer is operated in a positive, multireaction monitoring mode for the detection of TAL and AHP. The monitored transitions are 127–43.1 and 169–127 $m/z$ for TAL and AHP, respectively. For TAL quantification, a standard curve is prepared using >97% purity TAL in culture media ($R^2 > 0.99$). Recovery samples for TAL are tested regularly throughout analysis (average is 105%, range: 99–108%). As no genuine standard is available for AHP, relative peak integration values are used for comparison.

**Reporting summary**. Further information on research design is available in the Nature Research Reporting Summary linked to this article.

## Data availability
The raw MALDI-MS data that support the findings of this study are publicly available via Zenodo (https://doi.org/10.5281/zenodo.5601763). The authors declare that all other data supporting the findings of this study are available within the paper and its Supplementary information files. Source data are provided with this paper.

## Code availability
The bioinformatic pipeline is available on GitHub at https://github.com/AbateLab/Mass_spec_image_UMAP.

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

## Acknowledgements

This work was supported by the Chan Zuckerberg Biohub, the National Institutes of Health (NIH) (Grant Nos. R01-EB019453-02 and R01-HG008978-01), Office of the Director of National Intelligence (ODNI) (IARPA FELIX Contract No. N66001-18-C-4507), United States Department of Defense, Defense Advanced Research Projects Agency (DARPA) (Agreement No. W911NF1920013 The content of the information does not necessarily reflect the position or the policy of the Government, and no official endorsement should be inferred.) and funded in part by federal funds from the Virology Surveillance and Diagnosis Branch, Influenza Division, Centers for Disease Control and Prevention, under BAA 75D301-19-R-67835 (Topic #6). E.M.P. was supported by NIH T32 EB005582. R.T.K. thanks to the NSF Grant CHE-1904146.

## Author contributions

L.X. and A.R.A. conceptualized the project. L.X. designed and performed the experiments. L.X., K.-C.C., L.L., C.M.P and H.S.P were involved in selecting, designing, and constructing the enzyme library. K.-C.C. prepared the library. L.X. prepared and processed the custom substrate and MALDI-MS. N.T. and D.S.C provided access to the Bruker MALDI and assisted in data acquisition. E.M.P. and R.T.K. assisted in the validation of selected mutants. L.X., K.-C.C., E.M.P., C.M. and A.R.A. analyzed and interpreted the obtained data. L.X., C.M. and A.R.A. wrote the manuscript. All authors read and agreed to the final work.

## Competing interests

The authors declare no competing interests.
