## [Peer Review File · Nature Communications]

REVIEWER COMMENTS

Reviewer #1 (Remarks to the Author):

The paper describes a novel indirect sensing method to infer product formation in the context of enzyme engineering that is based on alterations in the host metabolism which are quantified by mass spectrometry (imaging).

The presented method is a novel scalable approach to this problem and as such can be expected to be a valuable contribution to the biotech community. We appreciate the creative application of MSI to this end. The figures are clear and well presented. We do have some remarks regarding the methodology and references.

p. 2 – caption figure 1:

We do not understand why PCA is mentioned here and not in the methods section or anywhere else in the paper. Can the authors clarify?

p. 4 l. 98 – 100:

The authors refer to the application of UMAP in the context of RNASeq but a more relevant application in this context is the evaluation of UMAP in the context of MSI:

<https://pubs.acs.org/doi/10.1021/acs.analchem.8b05827>

A reference to the original UMAP paper is missing, given that the Seurat package only incorporates the UMAP algorithm, the reference to this paper should be included as well:

McInnes, L, Healy, J, UMAP: Uniform Manifold Approximation and Projection for Dimension Reduction, ArXiv e-prints 1802.03426, 2018

Methods section MSI data analysis: The filter threshold is iteratively relaxed to incrementally include more m/z peaks until the library strain can be separated into distinct clusters:

The authors have used UMAP more as a way of visualizing the data, rather than a dimensionality reduction approach as such. Have the authors tried to reduce the complete MSI dataset to 3 dimensions and visualise the resulting embedding in a 3D scatter plot and as a hyperspectral (RGB)

visualization according to the well coordinates? We believe this should be a first step to evaluate the data in a complete unsupervised setting and if needed to apply thresholding as a second step.

p4. l. 98: To achieve the clearest clusters, we select the top nineteen m/z values with the highest signal-to-noise ratio and inter-sample variance

1. Clearest clusters: as in our previous remark, we want to warn for the risk of bias given that when applying a clustering algorithm, the algorithm will always find clear clusters at a certain point. While in this case the results could be validated, it is important to take potential bias into account and evaluate the data in an unsupervised fashion as well.

2. What do the authors mean with the SNR? How was the noise level determined? This part seems to be missing from the methods section and the supplementary figure provides no explanation either. This should be clarified.

3. Same for inter-sample variance, the information and explanation/clarification is missing from the methods section.

Taking into account the data, associated methodology and potential significance we believe the paper to be suitable for publication in Nature Communications after minor revisions.

Reviewer #2 (Remarks to the Author):

Overall, this presents a thorough demonstration of a mass spectrometry and microfluidics-based approach to screening libraries of enzymatic activity to improve the rapidity with which design/build/test/learn cycles can be undergone in synthetic biology. This is a proof of concept paper and overall the methods seem solid although it remains a proof of concept on a relatively simple system and it could use some more context and justification.

A few points need to be addressed to improve the clarity and reproducibility of this manuscript:

1. The rationale for the expression system, for example the 2PS promoter choice, is not well detailed in the supplemental or main text. I would like to see construct maps so that this work can be more easily replicated along with more explicit explanations and rationale for expression choices. I would also recommend putting the codon optimized sequences in the supplementary information.

2. The rationale for the utility of TAL as a model system (I know there are several with regard to its utility as a precursor for a variety of bioproducts) is not even mentioned in the text which leaves a reader who is not intimately familiar with the field somewhat lost. I would recommend briefly contextualizing TAL and why it is a valuable model system.

3. Figure S1 being moved into the main text would help a lot with clarity and understanding TAL vs. AHP as well as the saturation mutagenesis as well as Figure 3 in the main text

Reviewer #3 (Remarks to the Author):

This is an exciting approach to provide improvements to the test portion of metabolic engineering. The work is well done and well described. Great job!

The ability to use cells as sensors is well known in some fields, but not exactly in this way for improving the speed of testing large numbers of cell variants for products (within the metabolic engineering field). The work is high profile and fits the journal. A few minor points would improve the presentation.

As a minor point, the authors claim the work provides a quantitative readout which really is not demonstrated (instead, they use a variety of clustering approaches which are distinct). While perhaps some of the enzyme products can be quantified (which is difficult but possible via MALDI), can this be related to the levels of enzyme product. This would require a number of tricky calibrations, especially when the enzymatic product is not directly observed, which they list as a strength.

A question: if the size of the array to be imaged increases, the length of time the array is within the vacuum system increases, and so both MALDI matrix and perhaps some of the metabolites will sublime in a time dependent manner. Has this been observed as it has in other literature examples of larger arrays imaged via MALDI MS? Given the molecular targets highlighted here, this may be problematic. Have they observed such effects and would they expect them if the array size increased to the number of imaged spots to the hundreds of thousands per slide they claim are possible?

It may be good to place this a little better into context of other MALDI MS applications of microarrays. For example, compare their approach to other cell-based microarrays probed via MALDI MS (10.1021/acs.est.1c01138, 10.1007/978-1-4939-9831-9_9). Moving from cells, there are several recent examples related to metabolic engineering: (10.1002/bit.27343, 10.1016/j.copbio.2021.01.010, 10.1021/jasms.1c00013).

REVIEWER COMMENTS

Reviewer #1 (Remarks to the Author):

The paper describes a novel indirect sensing method to infer product formation in the context of enzyme engineering that is based on alterations in the host metabolism which are quantified by mass spectrometry (imaging).

The presented method is a novel scalable approach to this problem and as such can be expected to be a valuable contribution to the biotech community. We appreciate the creative application of MSI to this end. The figures are clear and well presented. We do have some remarks regarding the methodology and references.

p. 2 – caption figure 1:

We do not understand why PCA is mentioned here and not in the methods section or anywhere else in the paper. Can the authors clarify?

Response: The reviewer is correct that in the caption we refer to the clustering method as PCA, when it is a UMAP, a type of PCA plot. To address the reviewer's concern and minimize confusion, we've amended the caption to refer to this as a UMAP.

p. 4 l. 98 – 100:

The authors refer to the application of UMAP in the context of RNASeq but a more relevant application in this context is the evaluation of UMAP in the context of MSI:

<https://pubs.acs.org/doi/10.1021/acs.analchem.8b05827>

A reference to the original UMAP paper is missing, given that the Seurat package only incorporates the UMAP algorithm, the reference to this paper should be included as well: McInnes, L, Healy, J, UMAP: Uniform Manifold Approximation and Projection for Dimension Reduction, ArXiv e-prints 1802.03426, 2018

Response: We thank the reviewer for highlighting these references. We agree that it would be good to include them as important precursors to our work. Therefore, we have done as the reviewer has requested and added these references to our manuscript.

Methods section MSI data analysis: The filter threshold is iteratively relaxed to incrementally include more m/z peaks until the library strain can be separated into distinct clusters: The authors have used UMAP more as a way of visualizing the data, rather than a dimensionality reduction approach as such. Have the authors tried to reduce the complete MSI dataset to 3 dimensions and visualise the resulting embedding in a 3D scatter plot and as a hyperspectral (RGB) visualization according to the well coordinates? We believe this should be a first step to evaluate the data in a complete unsupervised setting and if needed to apply thresholding as a second step.

Response: To address the reviewer's comment, we have done as suggested and generated the 3D UMAP using the complete MSI dataset. We also RGB colored the UMAP according to the distance of each well to the center of the slide and highlighted the origin of the wells, whether from reference colonies or library. The results show that indeed clusters can be visualized in three dimensions (Figure S10, Video S1, Video S2). While the clusters are noisier and the reference peaks not as well localized when all the data is included, the major trends are the same. Interestingly, from the 3D UMAP of all peaks, we identify a new cluster emerging from the TAL cluster (Figure S10 c and d). By performing a differential analysis, we determine that metabolite m/z 103 is highly differentially expressed in these clusters. Using the Yeast Metabolome Database (YMDB) (Ramirez-Gaona, Miguel, et al. "YMDB 2.0: a significantly expanded version of the yeast metabolome database." *Nucleic acids research* 45.D1 (2017): D440-D445.) we identify 19 potential identities for this metabolite, although none are known products of the 2PS enzyme. This indicates that, either, this metabolite is an unknown side product of the enzyme, or an ancillary product generated by the host metabolism in reaction to the activities of these variants. Other molecules with different m/z appear to be up and downregulated between these clusters.

Overall, we think the 3D visualization and inclusion of all the data in the UMAP is interesting and insightful. Thus, to address the reviewer's comment, we have added a new supplemental figure, table, and discussion to the main text detailing these findings. We are thankful to the reviewer for making this suggestion as we believe the discovery of this unexpected metabolite nicely illustrates the power of metabolic biosensing for novel activity discovery.

p4. l. 98: To achieve the clearest clusters, we select the top nineteen m/z values with the highest signal-to-noise ratio and inter-sample variance

1. Clearest clusters: as in our previous remark, we want to warn for the risk of bias given that

when applying a clustering algorithm, the algorithm will always find clear clusters at a certain point. While in this case the results could be validated, it is important to take potential bias into account and evaluate the data in an unsupervised fashion as well.

Response: This is an excellent comment and an important issue in the generation of dimensionally reduced plots. Indeed, as the reviewer remarks, the specific parameters used to generate the clusters can have a significant impact on the plot. As a result, numerous accepted methods have been developed, which is what we've used to generate our UMAP. However, other approaches are of potential value, including unsupervised methods as described by the reviewer. Therefore, to address the reviewer's comment, we've re-clustered the data using the complete MSI dataset which is unsupervised. As explained in the previous comment, although the clusters are less distinct, the overall trends hold and are clear (Figure S10). In addition to the new supplementary figure and information, we've also added a more detailed discussion on the method we've used to select the 19 peaks for the two-dimensional UMAP.

2. What do the authors mean with the SNR? How was the noise level determined? This part seems to be missing from the methods section and the supplementary figure provides no explanation either. This should be clarified.

Response: We used the Bruker FlexImaging software to set the signal to noise ratio (SNR), where the threshold is defined as the height of the mass peak above its baseline relative to three times the standard deviation of the noise. The noise level is determined based on the peak-to-peak variation of the baseline (Müller, Matthias, et al. "Limits for the detection of (poly-) phosphoinositides by matrix-assisted laser desorption and ionization time-of-flight mass spectrometry (MALDI-TOF MS)." Chemistry and physics of lipids 110.2 (2001): 151-164.). To address the reviewer's comment, we have done as asked and amended the manuscript to now provide a detailed explanation of how the SNR is defined, calculated, and used in the data analysis.

3. Same for inter-sample variance, the information and explanation/clarification is missing from the methods section.

Response: To address inter-sample variance, we normalize using the total ion count (TIC), as is common in the field (Wulff, Jacob E., and Matthew W. Mitchell. "A comparison of various normalization methods for LC/MS metabolomics data." Advances in Bioscience and Biotechnology 9.08 (2018): 339.). In addition, we print reference strains of known composition in the first row of each block of ten rows to identify potential bias across the substrate, and with which we do not observe such bias (Figure S7c). To address this

comment, we have amended the manuscript to provide additional discussion on the normalization approach we've implemented and the usage of the reference spots to identify spatial bias.

Taking into account the data, associated methodology and potential significance we believe the paper to be suitable for publication in Nature Communications after minor revisions.

Response: We are happy the reviewer liked our manuscript and deems it acceptable for publication in Nature Communications.

Reviewer #2 (Remarks to the Author):

Overall, this presents a thorough demonstration of a mass spectrometry and microfluidics-based approach to screening libraries of enzymatic activity to improve the rapidity with which design/build/test/learn cycles can be undergone in synthetic biology. This is a proof of concept paper and overall the methods seem solid although it remains a proof of concept on a relatively simple system and it could use some more context and justification.

A few points need to be addressed to improve the clarity and reproducibility of this manuscript:

1. The rationale for the expression system, for example the 2PS promoter choice, is not well detailed in the supplemental or main text. I would like to see construct maps so that this work can be more easily replicated along with more explicit explanations and rationale for expression choices. I would also recommend putting the codon optimized sequences in the supplementary information.

Response: We thank the reviewer for these suggestions. To address the comment, we have done as the reviewer has suggested and added all the information to the manuscript and supplemental information, including additional text, figures and data (Fig. S1, and Table S5).

2. The rationale for the utility of TAL as a model system (I know there are several with regard to its utility as a precursor for a variety of bioproducts) is not even mentioned in the text which leaves a reader who is not intimately familiar with the field somewhat lost. I would recommend briefly contextualizing TAL and why it is a valuable model system.

Response: We thank the reviewer for pointing out this issue. We also agree that the selection of TAL for this demonstration should be justified as, indeed, TAL is an important

precursor to the synthesis of many high value molecules and was selected specifically for this reason. To address the comment, we have amended the text to add the discussion requested by the reviewer.

3. Figure S1 being moved into the main text would help a lot with clarity and understanding TAL vs. AHP as well as the saturation mutagenesis as well as Figure 3 in the main text
Response: We agree that this figure would be valuable in the main text. To address the comment, we have amended the manuscript to do as the reviewer has asked and move this figure into the main text. We are happy with the result as we believe the additional information will make the experimental design and enzyme activity easier to understand.

Reviewer #3 (Remarks to the Author):

This is an exciting approach to provide improvements to the test portion of metabolic engineering. The work is well done and well described. Great job!

Response: We are happy the reviewer liked our manuscript.

The ability to use cells as sensors is well known in some fields, but not exactly in this way for improving the speed of testing large numbers of cell variants for products (within the metabolic engineering field). The work is high profile and fits the journal. A few minor points would improve the presentation.

As a minor point, the authors claim the work provides a quantitative readout which really is not demonstrated (instead, they use a variety of clustering approaches which are distinct). While perhaps some of the enzyme products can be quantified (which is difficult but possible via MALDI), can this be related to the levels of enzyme product. This would require a number of tricky calibrations, especially when the enzymatic product is not directly observed, which they list as a strength.

Response: We agree with the reviewer that the quantitiveness of MALDI MS requires careful calibration of the instrument for each metabolite of interest. Thus, we performed calibration experiments with a titration series of purified TAL, which is the expected main product of the 2PS enzyme of our study (Figure S6). We also add reference colonies with quantified TAL production to the first row of every ten rows to account for other forms of measurement bias (temporal, spatial). Nevertheless, as the reviewer rightly highlights, quantitiveness of other metabolites cannot be assured since they have not been calibrated

and, in any case, the indirect sensing of the technique is what's of prime utility with our method. Therefore, to address the reviewer's comment, we have amended the text to indicate quantitative measurements only for TAL.

A question: if the size of the array to be imaged increases, the length of time the array is within the vacuum system increases, and so both MALDI matrix and perhaps some of the metabolites will sublime in a time dependent manner. Has this been observed as it has in other literature examples of larger arrays imaged via MALDI MS? Given the molecular targets highlighted here, this may be problematic. Have they observed such effects and would they expect them if the array size increased to the number of imaged spots to the hundreds of thousands per slide they claim are possible?

Response: We agree with the reviewer that if the array size is too large, sublimation may induce systematic bias such that wells scanned early may exhibit different spectral intensities from ones scanned later, which should appear as a systematic bias across our array. Our MALDI instrument scans at 10 kHz per laser pulse (Bruker Rapiflex), allowing us to image the entire slide in ~3 hrs. Over this time, we do not observe spatial bias in the signal (Figure S7 c). To account for any minor bias that might exist, we implement total ion count (TIC) normalization. To address the comment, we have amended the text in the main text and method to add the discussion requested by the reviewer.

It may be good to place this a little better into context of other MALDI MS applications of microarrays. For example, compare their approach to other cell-based microarrays probed via MALDI MS ([10.1021/acs.est.1c01138](https://doi.org/10.1021/acs.est.1c01138), [10.1007/978-1-4939-9831-9_9](https://doi.org/10.1007/978-1-4939-9831-9_9)). Moving from cells, there are several recent examples related to metabolic engineering: ([10.1002/bit.27343](https://doi.org/10.1002/bit.27343), [10.1016/j.copbio.2021.01.010](https://doi.org/10.1016/j.copbio.2021.01.010), [10.1021/jasms.1c00013](https://doi.org/10.1021/jasms.1c00013)).

Response: We thank the reviewer for this suggestion. To address the comment, we have done as asked and provided the additional requested discussion and references to these manuscripts.

REVIEWERS' COMMENTS

Reviewer #1 (Remarks to the Author):

Our remarks have been taken into account and we believe the adapted paper to be improved and suitable for publication in Nature Communications.

Reviewer #2 (Remarks to the Author):

The changes have strengthened the manuscript. My one suggestion would be the following:

Line 77—could use a reference “TAL has been used as a platform precursor for synthesis of high value chemicals commonly 77 derived from fossil fuels.”

Reviewer #3 (Remarks to the Author):

The authors have done an effective job in responding to the reviewers. This is an exciting contribution to the literature.

REVIEWER COMMENTS

Reviewer #1 (Remarks to the Author):

Our remarks have been taken into account and we believe the adapted paper to be improved and suitable for publication in Nature Communications.

Response: We are pleased that the reviewer has found the revised manuscript suitable for publication, and once again thank the reviewer for their time evaluating our work.

Reviewer #2 (Remarks to the Author):

The changes have strengthened the manuscript. My one suggestion would be the following:

Line 77—could use a reference “TAL has been used as a platform precursor for synthesis of high value chemicals commonly derived from fossil fuels.”

Response: We thank the reviewer for this helpful suggestion. References are indeed needed for this statement, and we have added a citation to this sentence in the main text.

Reviewer #3 (Remarks to the Author):

The authors have done an effective job in responding to the reviewers. This is an exciting contribution to the literature.

Response: We are pleased that the reviewer has found the revised manuscript suitable for publication, and once again thank the reviewer for their time evaluating our work.